# HD-KT: Advancing Robust Knowledge Tracing via Anomalous Learning Interaction Detection

Submitted for Blind Review

## ABSTRACT

Knowledge tracing (KT) is a crucial task in online learning, aimed at tracing and predicting each student's knowledge states throughout their learning process. Over the past decade, it has garnered widespread attention due to it provides the potential for more tailored and adaptive online learning experiences. Although most current KT methodologies emphasize optimizing network structures to enhance predictive accuracy for future student performance, they often neglect anomalous interactions in students' learning processes, which may arise from low data quality (i.e., inferior question quality) and abnormal student behaviors (i.e., guessing and mistakes). To this end, in this paper, we propose a novel framework, termed **HD-KT**, designed to enhance the robustness of existing **KT** methodologies with **H**ybrid learning interactions **D**enoising approach. Specifically, we introduce two detectors for anomalous learning interactions, namely knowledge state-guided anomaly detector and student profile-guided anomaly detector. In the first detection module, we design a sequential autoencoder to identify anomalous learning interactions by detecting atypical student knowledge states. In the second module, we incorporate an attention mechanism by modeling a student's long-term profile to capture irregular interactions. Extensive experiments on four real-world benchmark datasets have decisively shown our HD-KT markedly boosts the robustness of numerous prevailing KT models, consequently increasing the accuracy of future student performance predictions. Additionally, our case studies highlight the versatility of HD-KT in addressing diverse downstream tasks, such as exercise quality analysis and learning behavior-based student clustering.

## KEYWORDS

Intelligent education, online learning, knowledge tracing, anomaly detection

## 1 INTRODUCTION

In recent years, online learning has seen significant growth [28], delivering substantial assistance to educators in their teaching methods and empowering students in their learning journeys [27, 36]. Knowledge tracking (KT) stands as a pivotal task in online learning, focusing on tracing students' knowledge states through their sequential exercises on various knowledge concepts, ultimately

*WWW'24, May 13-17, 2024, Singapore*
© 2023 Association for Computing Machinery.
ACM ISBN 978-x-xxxx-xxxx-x/YY/MM. . . $15.00
https://doi.org/10.1145/nnnnnnn.nnnnnnn

enabling us to predict their future performance [6, 34, 44]. Consequently, online learning systems employ KT to provide educators and students with a comprehensive understanding of their strengths and weaknesses in mastering knowledge, as well as the patterns in students' learning behaviors. This, in turn, enables the delivery of more tailored and adaptive online learning services [24].

In the realm of KT, the pivotal aspect is modeling the learning interaction sequences between students and exercises to capture the evolving states of student knowledge. Traditional approaches, such as *Bayesian Knowledge Tracing* (BKT) [6] and its variants [51, 16], employ the *Hidden Markov Process* for this purpose. Over the past decade, with the rapid advancements in deep learning, neural KT models (KTMs) utilizing architectures like recurrent neural networks (RNNs) and transformers have been introduced, elevating the efficacy of KT [35, 17, 15]. While most current approaches prioritize optimizing network structures to boost the predictive accuracy of future student performance, they frequently overlook the anomalous interactions in students' learning processes, thereby compromising the reliability of inferred students' knowledge states. Such anomalous learning interactions might stem from low question quality or abnormal student behaviors. For instance, as illustrated in Figure 1, when a student presents two conflicting responses for exercise $e_6$ over a short period, it is reasonable to deduce that one of these interactions constitutes data noise. Meanwhile, at the 5-th moment, as diagnosed by the KTM, the student exhibits a high mastery level on *"Square Root"*, however, he/she mistakenly answers exercise $e_4$ incorrectly. This anomalous learning interaction could be attributed to either the student's inadvertence or the poor quality of exercise $e_4$, subsequently resulting in an inaccurate inference regarding the student's state of this particular concept. To delve deeper into the impact of anomalous interactions on KTMs, we introduced random noise into the ASSISTment12 dataset [8] by randomly learning interactions and inverting the corresponding responses. Subsequently, we employed the DKT method to infer students' knowledge proficiency and predict their future performance at the next time step. As shown in Figure 2, with the rise of anomalous data, there is a notable increase in the variability of students' proficiency states and the AUC metric declines by 2.04%.

Indeed, identifying anomalies within students' learning interactions and subsequently elevating the performance of KTMs presents a significant challenge, particularly in light of our absence of historically labeled anomalous data. To this end, in this paper, we propose a novel framework, namely **HD-KT**, designed to enhance the robustness of existing **KT**Ms with **H**ybrid learning interactions **D**enoising approach. Specifically, we start with an embedding layer to learn the representations for students, exercises, and concepts based on the student's learning sequence. Then, we design two novel detectors for anomalous learning interactions, namely knowledge state-guided anomaly detector and student profile-guided anomaly detector. In the first detector, a sequential variational autoencoder is crafted to identify anomalous learning interactions by

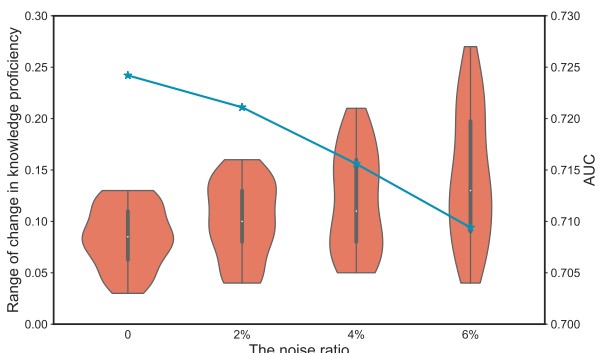

| Exercises | Knowledge Concepts |
|-----------|--------------------|
| $e_1$ | $c_1$: Square Root |
| $e_2$ | $c_2$: Unit Rate |
| $e_3$ | $c_3$: Proportion |
| $e_4$ | $c_1$: Square Root |
| $e_5$ | $c_4$: Multiplication Integers |
| $e_6$ | $c_5$: Distributive Property |

**Figure 1: The illustrative examples of a sequence of interactions for a student learning online and the corresponding diagnosed knowledge states. The record comprises 9 learning interactions, spanning 6 exercises and encompassing 5 knowledge concepts.**

**Figure 2: The impact of different proportions of noise data on the DKT method [35] in the ASSISTment12 dataset. The range of change in knowledge proficiency denotes the maximum rate of change in each student's mastery level for different knowledge concepts within a learning process.**

detecting atypical student knowledge states. In the second detector, we present an effective attention mechanism, integrated with modeling students' long-term characteristics, to capture irregular interactions. In particular, both modules modeling different aspects of anomaly perception are jointly exploited to denoise and refine the sequential exercising behaviors of learners. Subsequently, we introduce the KTM adaptor, which allows the integration of different KT models is conducted to realize the prediction of the future response performance of students. Extensive experiments on four real-world benchmark datasets have decisively shown HD-KT markedly boosts the robustness of numerous prevailing KT models, consequently increasing the accuracy of future student performance predictions. Additionally, our case studies highlight the versatility of HD-KT in addressing diverse downstream tasks, such as exercise quality analysis and learning behavior-based student clustering.

## 2 RELATED WORK

### 2.1 Knowledge Tracing

Researchers have explored different modes of KT conduction. Existing KT methods can be divided into two types: probabilistic or logistic model-based traditional methods and deep learning-based methods. Probabilistic model-based methods generally define a student's knowledge states as a binary variable and use *Hidden Markov Model* to estimate the student's conceptual mastery level, and the representatives include BKT [6] and its variants [14]. Logistic model-based methods mainly estimate student performance

by usually learning a logistic function, based on different factors in some students who solve the same set of problems, and the representatives include *Performance Factor Analysis* [33] and *Learning Factor Analysis* [4]. Differently, deep learning-based KT methods leverage various neural network techniques to solve the sequential prediction task of the student answering exercises for tracing the student's knowledge states, which are commonly implicit in the hidden states of models. The representatives include the RNN-based method DKT [35], memory-augmented methods (e.g., DKVMN [53]), attention mechanism-based methods [29, 9], transformer-based methods [17, 15] and graph neural network-based methods [40, 39, 46].

Among them, some KT methods not only focus on designing novel network architectures but also try to solve some intrinsic difficulties in KT. For example, CL4KT [20] and CMKT [25] aim to address the student-exercise interaction sparseness problem; ATKT [10] and DLKT [12] pursue to improve model generalization performance; LPKT [38], HawkesKT [42] and CT-NCM [26] attempt to model the forgetting behaviors of students during the learning process; DTransformer [48] was proposed to obtain stable knowledge state estimation and tracing, instead of only improving the prediction performance, by inventing a new training paradigm. It can be observed that many intrinsic difficulties (including sparseness, forgetting, stable tracing, and so on) in KT have been well solved, but how to overcome the influence caused by the abnormal conditions that occur among students during the online learning process has been less explored. The abnormal conditions may arise from low data quality and abnormal student behaviors, which is ubiquitous in online learning system and will affect the accuracy and interpretability of KT tasks, and thus it is urgent for us to develop corresponding KT methods to solve this difficulty.

### 2.2 Anomaly Detection

Anomaly detection is an important research topic with broad application prospects. For example, in the recommendation system, there are certain abnormal behaviors in the user's click sequence (such as clicking on a product that he does not like), which will affect the recommendation of the next item for the user [55, 50]. In industry, researchers detect whether abnormalities occur in the sensors to improve production efficiency [3, 37, 30]. Anomaly detection has been applied for various types of data. Here we focus on anomaly detection for time series data. These existing researches can be divided into prediction-based methods [7, 54] and reconstruction-based methods [21, 41]. Prediction-based models utilize advanced

machine learning components to predict the future variable performance based on the historical time series through modeling the spatiotemporal correlation between variables in time series data. The abnormality is detected through prediction probabilities. In order to improve the accuracy of abnormality detection, a variety of discriminant models attempt to better learn the complex relationship between variables to enhance the prediction performance. For example, *Deng and Hooi* [7] proposed a graph neural network based prediction model to capture complex inter-sensor relationships to detect and explain anomalies that deviate from these relationships. *Zhao et al.* [54] combined feature-oriented graph attention network (GAT) and time-oriented GAT to handle spatial dependence and temporal dependence in predicting. Reconstruction-based methods pursue precise representations of the entire time series data for data reconstruction, and detect anomalies according to the difficulty of reconstruction. To be specific, it is more difficult to reconstruct abnormal data and less difficult to reconstruct normal data. Therefore, this category pursues to learn robust and accurate representations of input data for reconstructing input data. For example, *Li et al.* [21] used the generative adversarial network (GAN) framework with long short-term memory (LSTM) as the basic unit to accurately reconstruct input data by considering the entire set of variables concurrently. In the literature [41], the proposed OmniAnomaly uses stochastic recurrent neural networks (RNN) to find robust representations for multivariate time series. *Audibert et al.* [2] proposed an AutoEncoder architecture with adversarial learning inspired by GANs. Recent work [1] exploits spectral analysis of latent representations and produces simultaneous representations of multivariate data. However, to the best of our knowledge, no researchers have head-on addressed the anomaly issue in knowledge tracing tasks.

## 3 PROBLEM DEFINITION

In this section, we formally define the problem of knowledge tracing (KT). Suppose there is a set of $N$ students, $\mathcal{S} = \{s_1, s_2, \ldots, s_N\}$, a set of $M$ exercises, $\mathcal{E} = \{e_1, e_2, \ldots, e_M\}$, and s set of $C$ knowledge concepts, $\mathcal{K} = \{k_1, k_2, \ldots, k_C\}$. Each exercise is associated with specific knowledge concepts and the $Q$-matrix $Q = \{q_{ij} \in \{0, 1\}\}^{M \times C}$ is utilized to indicate the relationship between exercises and knowledge concepts, where $q_{ij} = 1$ if exercise $e_i$ involves concept $k_j$ and $q_{ij} = 0$ otherwise. For the exercise-solving sequence for each student during the learning process, we denote it with $\mathcal{R} = \{(e_1, k_1, r_1), (e_2, k_2, r_2), \ldots, (e_T, k_T, r_T)\}$, where the triplet $(e_t, k_t, r_t)$ is the $t$-th learning interaction behavior, and $e_t \in \mathcal{E}$, $k_t \in \mathcal{K}, r_t \in \{0, 1\}$ represent the answered question, the related knowledge concept and the response result, respectively.

**Problem Definition.** *Given students' learning sequence $\mathcal{R} = \{(e_1, k_1, r_1), (e_2, k_2, r_2), \ldots, (e_T, k_T, r_T)\}$, the KT task aims to monitor students' evolving knowledge state during the learning process and predict their future performance at the next time step $T + 1$, which can be further applied to individualize students' learning scheme and maximize their learning efficiency.*

## 4 METHODOLOGY

In this section, we initially provide an overall overview of our proposed framework **HD-KT** (short for **H**ybrid learning interactions **D**enoising **K**nowledge **T**racing). Subsequently, we explore each component of the model with a detailed explanation.

**Overview.** Our HD-KT model innovatively introduces the measurement of anomalous factors during the students' learning processes, effectively achieving robust knowledge tracing through the implementation of the hybrid learning interaction denoising strategy. As shown in Figure 3, the overall architecture of HD-KT consists of four main components, including the embedding layer, the knowledge state-guided anomaly detector, the student profile-guided anomaly detector, and the KTM adaptor. Specifically, by taking learning sequence, the embedding layer first outputs the vectorized representation of students, exercises and concepts. In the first detector, a knowledge concept-aware sequential variational autoencoder is designed to reconstruct the proficiency distribution of students with the dimension of knowledge concepts. Meanwhile, we leverage an effective attention mechanism with modeling students' long-term characteristics to explore anomalous interactions in the student profile-guided anomaly detector. In particular, both of these signals modeling different aspects of anomaly perception are jointly exploited to denoise and refine the sequential exercising behaviors of learners. Finally, the KTM adaptor that allows the integration of different KT models is conducted to realize the prediction of the future response performance of students.

### 4.1 Embedding Layer

As is well known, the learning process of students is inherently intricate, characterized by students progressively engaging with exercises and continually enhancing their cognitive abilities [6, 35]. In HD-KT, to effectively model the response interaction behaviors during the learning process of students, we consider the following elements: students, exercises, concepts, answers, and knowledge status. We define the basic unit of the learning process as the triplet *exercise-concept-response* and construct an embedding layer to encode them with trainable parameter matrices. Specifically, for the $t$-th exercising behavior $(e_t, k_t, r_t)$ of student $s$, we transform them into the corresponding embedded representations by multiplying their one-hot vectors with the parameter matrices:

$$\mathbf{x}_{e_t} = \mathbf{e}_t W^E, \ \mathbf{x}_{a_t} = \mathbf{a}_t W^A, \tag{1}$$

where $\mathbf{e}_t \in \mathbb{R}^M$ and $\mathbf{a}_t \in \mathbb{R}^{2C}$ denote the one-hot vector of the exercise and the response interaction, respectively; $\mathbf{x}_{e_t} \in \mathbb{R}^{d_e}$ and $\mathbf{x}_{a_t} \in \mathbb{R}^{d_a}$ stand for their embeddings representations; $W^E \in \mathbb{R}^{M \times d_e}$ and $W^A \in \mathbb{R}^{2C \times d_a}$ denote the trainable weight matrices; $d_e$ and $d_a$ are corresponding dimensions. In particular, $\mathbf{a}_t$ here is the response interaction vector representing the knowledge performance, which is obtained by combining the knowledge concept $c_t$ and the answer $r_t$:

$$\mathbf{a}_{t,i} = \begin{cases} 1, i = k_t + C \cdot r_t \\ 0, otherwise \end{cases} \tag{2}$$

Furthermore, we introduce an adaptable embedding representation $\mathbf{x}_s = \mathbf{s} W^S$ for student $s$ to delineate its profile, which supports the consistency of knowledge evolution, thus facilitating the exploration of the learning trajectory, where $\mathbf{s} \in \mathbb{R}^N$ denotes the one-hot vector of student $s$, $\mathbf{x}_s \in \mathbb{R}^{d_s}$ is the global student profile, $W^S \in \mathbb{R}^{N \times d_s}$ denotes the trainable weight matrix, and $d_s$ is the corresponding embedding size. Finally, to effectively model each of the student's learning behaviors, with reference to [38], we acquire the learning embedding by fusing the exercise representation and the knowledge performance representation together and employing

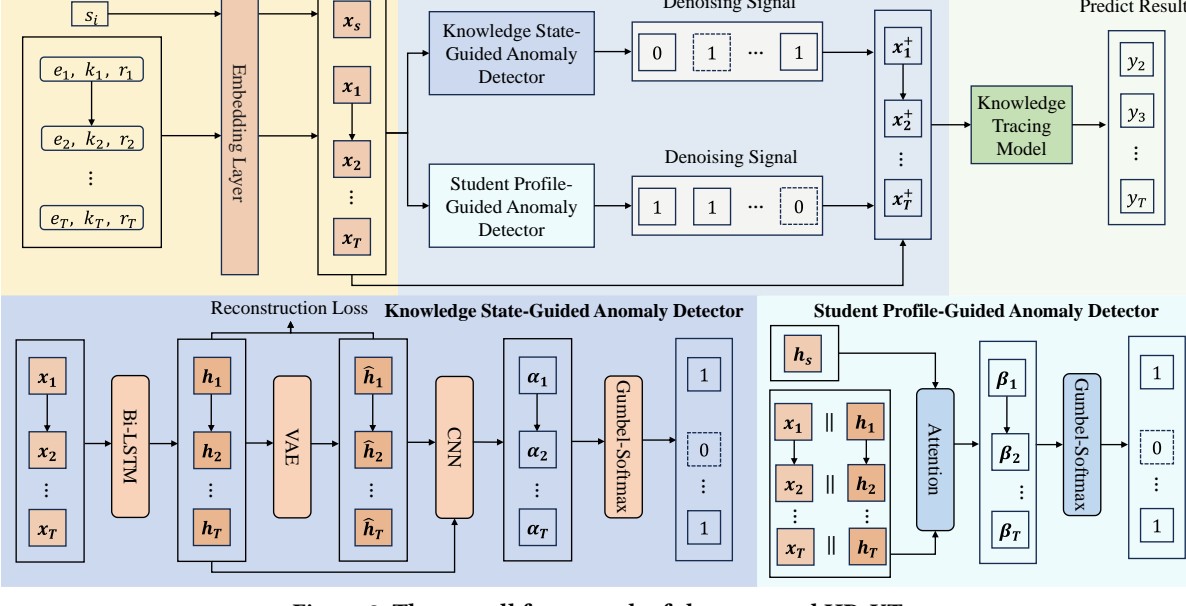

**Figure 3: The overall framework of the proposed HD-KT.**

a multi-layer perceptron (MLP) as follows:

$$\mathbf{x}_t = [\mathbf{x}_{e_t} \| \mathbf{x}_{a_t}] W_1 + b_1, \tag{3}$$

where $\|$ denotes the operation of concatenating, $W_1 \in \mathbb{R}^{(d_e+d_a) \times d}$ is the weight matrix, $b_1 \in \mathbb{R}^d$ is the bias term, $d$ is the dimension. As a result, we get the representation of the learning sequence of student $s$: $\mathbf{X}_s = [\mathbf{x}_1, \mathbf{x}_2, \ldots, \mathbf{x}_T] \in \mathbb{R}^{T \times d}$.

## 4.2 Knowledge State-Guided Anomaly Detector

The consistency and gradual progression of competence growth are recognized as inherent characteristics of the student's learning process [38, 47]. Nonetheless, in real and intricate learning environments, anomalous signals can manifest due to external influences, e.g., a student correctly answers multiple questions that he has not genuinely mastered, potentially due to cheating, or a highly proficient student may inaccurately respond to straightforward exercises due to carelessness, among other possibilities. Therefore, in this part, we develop a knowledge state-guided anomaly detector to explore the anomalous signals thus enabling more effective modeling and diagnosing of student learning behaviors.

Firstly, to proficiently exploit the sequential learning behaviors of students and capture dependencies in the contextual knowledge states, the encoded bidirectional long short-term memory network (Bi-LSTM) [13] is utilized to model and process the embedded learning sequence representation $\mathbf{X}^s$ as follows:

$$\begin{aligned} \tilde{\mathbf{H}}_s^L, \tilde{\mathbf{H}}_s^R &= Bi\text{-}LSTM(\mathbf{X}_s, \Theta_1), \\ \mathbf{H}_s &= \mathbf{H}_s^L \oplus \mathbf{H}_s^R, \end{aligned} \tag{4}$$

where $\tilde{\mathbf{H}}_s^L, \tilde{\mathbf{H}}_s^R \in \mathbb{R}^{T \times d_s}$ represent the bidirectional intermediate hidden states, respectively, $\mathbf{H}_s = [\mathbf{h}_1, \mathbf{h}_2, \ldots, \mathbf{h}_T] \in \mathbb{R}^{T \times d_s}$ denotes the knowledge state matrix, $Bi\text{-}LSTM(\cdot)$ refers to the Bi-LSTM network architecture, $\Theta_1$ is the corresponding trainable parameterset, and $\oplus$ stands for the element-wise addition operator. After obtaining the student knowledge states, inspired by [23], we contemplate

utilizing a *Variational Autoencoder* (VAE[11]) to reconstruct the temporal evolving competencies for capturing anomalous signals during the learning process. Specifically, we model the latent variable $\hat{\mathbf{H}}_s$ to adhere to a Gaussian distribution for deriving more robust embedding as follows:

$$\hat{\mathbf{H}}_s \sim \mathcal{N}(\mu, \sigma^2), \ \mu = MLP_\mu(\mathbf{H}_s), \ \sigma = MLP_\sigma(\mathbf{H}_s), \tag{5}$$

where $\hat{\mathbf{H}}_s = [\hat{\mathbf{h}}_1, \hat{\mathbf{h}}_2, \ldots, \hat{\mathbf{h}}_T] \in \mathbb{R}^{T \times d_s}$ represents the reconstructed sequential competency level consisting of the knowledge state at each time step, and both $MLP_\mu(\cdot)$ and $MLP_\sigma(\cdot)$ are two trainable MLP networks for learning the distribution parameters. After getting the reconstructed knowledge state sequence, we can calculate the completed reconstruction loss as follows:

$$\begin{aligned} \mathcal{L}^{Rec} &= \frac{1}{T} \sum_{t=1}^{T} (\hat{\mathbf{h}}_t - \mathbf{h}_t)^2 + \mathcal{L}^{kl}, \\ \mathcal{L}^{kl} &= \sum_{1 \le t \le T} \mu_t^2 + \sigma_t^2 - log(\sigma_t). \end{aligned} \tag{6}$$

After acquiring the reconstructed student knowledge state, intuitively, we can capture the inconsistency by integrating it with the student's initial knowledge level and inputting this combined information into the fully connected layers. Nevertheless, the minimization of the reconstruction loss can make it challenging to discern the distinctions between the aforementioned representations. Inspired by previous works [22, 52], we endeavor to leverage a convolutional neural network (CNN) to enhance the detection capacity for capturing disparities among distinct representations of the same dimension. Specifically, we concatenate the original embedding of the knowledge state with the decoded representation at each moment and utilize a convolution operator to preserve dimensional information by:

$$\begin{aligned} \alpha_t &= \sigma(\mathbf{C}_t W_2), \\ \mathbf{C}_t &= Conv([\hat{\mathbf{h}}_t \| \mathbf{h}_t], \Theta_2), \end{aligned} \tag{7}$$

where $\mathbf{C}_t \in \mathbb{R}^{d_s}$ is the output of the convolution layer, $conv(\cdot)$ is a two-dimensional convolution operation with a filter size of 2×1 and a stride of 1, $\Theta_2$ is the trainable parameter of each channel, $W_2 \in \mathbb{R}^{d_s \times 2}$ is the trainable parameter matrix. Notably, $\boldsymbol{\alpha}_t \in \mathbb{R}^2$ denotes the relation vector, where the first dimension represents the consistency between $\hat{\mathbf{h}}_t$ and $\mathbf{h}_t$, while the second dimension refers to the inconsistency. Therefore, the scores can be treated as a binary distribution (i.e., consistency vs. inconsistency). To generate binary values (i.e., 0 vs. 1) and facilitate gradient back-propagation, we utilize a Gumbel-Softmax function [43, 45, 49] to support the learning of model via:

$$\hat{\boldsymbol{\alpha}}_t = Gumbel\text{-}Softmax(\boldsymbol{\alpha}_t, \tau),$$
$$= \frac{exp(\log(\boldsymbol{\alpha}_{t,i}) + g_i)/\tau}{\sum_{j=0}^{1} exp(\log(\boldsymbol{\alpha}_{t,j}) + g_j)/\tau}. \quad (8)$$

where $\hat{\boldsymbol{\alpha}}_t \in \mathbb{R}^2$ denotes whether the changes in student's knowledge status is stable, $g_j$ is i.i.d sampled from the Gumbel distribution as noise disturber, $Gumbel\text{-}Softmax(\cdot)$ denotes the Gumbel-Softmax function and $\tau > 0$ is the temperature parameter that controls the selection distribution. When $\tau \longrightarrow 0$, $\hat{\boldsymbol{\alpha}}_t$ approximates a one-hot vector (i.e. hard selection). When $\tau \longrightarrow \infty$, $\hat{\boldsymbol{\alpha}}_t$ approximates a uniform distribution. When $\tau \longrightarrow 1$, the Gumbel-Softmax function is the same as the general Softmax function.

## 4.3 Student Profile-Guided Anomaly Detector

Due to the unique attributes of each student within the learning process, even when subjected to the same learning experience, differential learning outcomes and memory retention may be observed. We contend that this phenomenon imparts crucial insights into sequence denoising, specifically, the fact that abnormal learning behaviors frequently exhibit substantial deviations from the individual characteristics of students throughout the learning process. Therefore, we design a student profile-guided anomaly detection module to explore the asymptotic smoothness of the student's evolving competency. Specifically, we develop an attention module as the discriminator to detect inconsistency between the learning status and the student profile, which utilizes the student representation as a query vector and assign different attention weight to each learning encoding within the learning sequence:

$$\boldsymbol{\beta}_t = \sigma(tanh([\mathbf{x}_t \| \mathbf{h}_t]W_3 + \mathbf{h}_s W_4)W_5), \quad (9)$$

where $\boldsymbol{\beta}_t \in \mathbb{R}^2$ is the $t$-th attention vector, $W_3 \in \mathbb{R}^{2d_s \times d}$, $W_4 \in \mathbb{R}^{d_s \times d}$ and $W_5 \in \mathbb{R}^{d \times 2}$ are the trainable parameter matrix, and $\sigma(\cdot)$ and $tanh(\cdot)$ denote the sigmoid and tanh activation function, respectively. Notably, the first dimension of $\boldsymbol{\beta}_t$ represents the consistency between student response performance and student learning profile, as well as the second dimension denotes the inconsistency. Therefore, the scores can be viewed as binary distributions (i.e., consistency vs. inconsistency), and then we leverage a similar process to generate binary value for $\boldsymbol{\beta}_t$ via:

$$\hat{\boldsymbol{\beta}}_t = Gumbel\text{-}Softmax(\boldsymbol{\beta}_t, \tau),$$
$$= \frac{exp(\log(\boldsymbol{\beta}_{t,i}) + g_i)/\tau}{\sum_{j=0}^{1} exp(\log(\boldsymbol{\beta}_{t,j}) + g_j)/\tau}, \quad (10)$$

where $\hat{\boldsymbol{\beta}}_t \in \mathbb{R}^2$ denotes the predicted anomalous vector about the knowledge state of student $s$, and $\tau$ is the same temperature

parameter used in formula Eq. (8) to tune the learned distribution from the Gumbel-Softmax function.

## 4.4 KTM Adaptor

With the previously mentioned anomaly detectors, the proposed HD-KT model enables to detect noise components within the sequence based on signals derived from the knowledge state and student profile levels, which involves labeling a response as noise when it exhibits inconsistency with the respective student attributes or the amalgamated knowledge state. Nevertheless, in practical applications, the false positives may be introduced, leading to the inadvertent exclusion of valuable information essential for predicting student performance. Hence, we advocate the development of a more stringent criterion for the elimination of anomalous learning interaction, aimed at retaining solely dependable noise-free data while preserving valuable information. An instance is categorized as noise only when incongruities are concurrently identified in both signals, typically adhering to the principle of consensus. Formally, we generate noise-free sequences from the input sequential learning behaviors of individual KTM via the following steps:

$$p_t = 1 - a_t \times b_t, \quad (11)$$

$$\mathbf{X}_s^+ = [p_1\mathbf{x}_1, p_2\mathbf{x}_2, \ldots, p_T\mathbf{x}_T], \quad (12)$$

where $p_t \in \{0, 1\}$ indicates whether an learning interaction is noisy (i.e., $p_t = 0$) or not, $a_t$ and $b_t$ denote the second dimension scalar of above mentioned $\boldsymbol{\alpha}_t$ and $\boldsymbol{\beta}_t$, respectively. Note that we apply the denoised signal to the embedding representation of learning sequence $\mathbf{X}_s$ to support the gradient backpropagation. Particularly, we design a KTM adaptor to adapt our proposed HD-KT framework for the integration into various mainstream knowledge tracing model for predicting the feature response performance of students, and we formalize as follows:

$$\hat{y} = KTM(\mathbf{X}_s^+), \quad (13)$$

where KTM is a basic knowledge tracing model (e.g., DKT, LPKT, etc.), which takes the denoised learning sequence representation $\mathbf{X}_s^+$ as input, and outputs the predicted future performance $\hat{y}$.

## 4.5 Model Optimization

In the training phase, we mainly evaluate the performance of the predicted student's responses in the interaction sequences. Similar to [38, 35], the binary cross entropy loss function between the predicted value $\hat{y}_t$ of student $s$ at time step $t$ and the ground truth $r_t$ is utilized, as follows:

$$\mathcal{L}^{Pre} = -\sum_{t=1}^{T} (r_t \log(\hat{y}_t) + (1 - r_t) \log(1 - \hat{y}_t)). \quad (14)$$

where $\mathcal{L}^{Pre}$ represents the prediction loss. Meanwhile, we also introduce the reconstruct loss to enhance the stability of parameter training of the anomaly detector according to Eq. (6), and build the final training loss as follows:

$$\mathcal{L} = \frac{1}{|\mathcal{S}|} \sum_{s \in \mathcal{S}} (\mathcal{L}_s^{Pre} + \mathcal{L}_s^{Rec}) + \lambda \|\Theta\|_2^2, \quad (15)$$

where $\lambda$ represents the hyperparameter of $L_2$ regularization strength, and $\Theta$ is the set of all model parameters. The objective function was minimized using Adam optimizer [18] on mini-batches. More details of settings are specified in the part of experiments.

**Table 1: Statistics of all datasets.**

| Dataset | #Students | #Concepts | #Exercises | #Interactions |
|---|---|---|---|---|
| ASSISTment12 | 25.3k | 245 | 50.9k | 2,621.3k |
| ASSISTment17 | 1.7k | 102 | 3.2k | 942.8k |
| Slepemapy.cz | 81.7k | 1,458 | 2.9k | 9,786.5k |
| Junyi | 175.4k | 40 | 0.7k | 25,670.2k |

## 5 EXPERIMENT

In this section, we conduct a series of experiments using four real-world benchmark datasets to validate the efficacy of our proposed model. We aim to address the subsequent research questions:

- **RQ1**: Can our proposed HD-KT framework effectively enhance the performance and robustness of the existing KT models?
- **RQ2**: What benefit does each component of the proposed HD-KT model offer?
- **RQ3**: Does our approach facilitate the analysis of question quality and enable student clustering based on learning behaviors?

## 5.1 Experimental Setting

*5.1.1 Datasets.* In this paper, we conducted our experiments on four public benchmark datasets, i.e., ASSISTment12, ASSISTment17, Slepemapy.cz, and Junyi. The ASSISTment12 dataset, referenced in [8], was collected from the ASSISTments online tutoring system and encompasses student activity data for the academic year 2012-2013. ASSISTment17 [32] was released during the ASSISTments Longitudinal Data Mining Competition in 2017. The dataset Slepemapy.cz [31] originates from an online adaptive system, i.e., slepemapy.cz, for practicing geography. The Junyi dataset [5] was collected from the Junyi Academy, an E-learning platform established in 2012. To optimize calculation efficiency, we followed [38] to set the maximum sequence length to 50 and truncate the learning sequences exceeding this length into multiple sub-sequences. To ensure reasonableness, we screened out the sequences with lengths less than 5. The statistics of four datasets are shown in Table 1.

*5.1.2 Evaluation Metrics.* We employed both accuracy (ACC) and the area under the receiver operating characteristics curve (AUC) as metrics to assess the efficacy of various methods in predicting the binary outcomes of future student responses to exercises.

*5.1.3 Baseline Methods.* To validate that our proposed HD-KT framework can significantly enhance the performance of different KT models, we selected three representative KT models as the backbone, including DKT, HawkesKT, and LPKT. The details are displayed as follows:

- **DKT** [35] pioneered the use of Recurrent Neural Networks (RNNs) to model students' knowledge states, inferring current exercise performance from past learning records. In our implementation, we employed the LSTM architecture.
- **HawkesKT** [42] posits that students' proficiency in each knowledge concept is influenced not only by prior interactions with that concept but also by other relevant concepts, termed as cross-effects among knowledge concepts. HawkesKT employs collaborative filtering and matrix factorization to discover the temporal cross-effects between different concepts.

- **LPKT** [38] distinguishes students' absorption of knowledge and forgetting of knowledge during the learning process through specially designed learning gates and forgetting gates respectively. The state undergoes intermittent updates via a straightforward weighted blend of both learning and forgetting factors.

We applied our framework to these models, resulting in three variants named HD-DKT, HD-HawkesKT, and HD-LPKT. Additionally, we selected two representative anomaly information section methods in the field of time series and sequential recommendation to serve as the baselines, including:

- **DSAN** [50], known as the dual sparse attention network, is designed to pinpoint items in a recommendation system that diverge from the user's anticipated preferences by assigning unique weights to each item in the sequence. In our experiment, we treated students and exercises as target item and interactive items, respectively. By integrating DSAN with various KT models, we leveraged the unique weights within DSAN to detect anomalous data.
- **GDN** [7], known as the graph deviation network, is a prediction-based multivariate temporal anomaly detection method leveraging graph attention (GAT) to capture the relationships within each feature of the time series data. In our experiments, we treat the sequential knowledge states on different concepts as the multivariate time series.

Moreover, we compared our HD-KT with the state-of-the-art robust KT model, that is,

- **DTransformer** [48], which introduces a unique transformer-based architecture combined with a novel training paradigm to achieve consistent and reliable knowledge state tracing.

*5.1.4 Implementation Details.* In our experiment, we performed 5-fold cross-validation. Specifically, the 80% of the learning sequences are split as the training set (70%) and the validation set (10%), while the rest 20% are used as the test set. We faithfully implemented DKT, HawkesKT and LPKT based on their original papers. To be specific, if parameters were consistent across various datasets in the original paper, we retained them as described (e.g., all parameters for HawkesKT). However, if the sensitivity to datasets was indicated, we performed parameter tuning on the validation set, adhering to the value ranges specified in the original works (e.g., parameters for DKT). We performed hyperparameter tuning for each KTM combined with HD based on the validation set. We searched the embedding size in [16, 32, 64, 128], hidden size in [16, 32, 64, 128], and dropout rate in [0, 0.1, 0.2, 0.25]. We used the Adam algorithm [19] as the optimizer. All experiments were implemented with PyTorch by Python and conducted with GeForce RTX4090 GPU.

## 5.2 Overall Performance (RQ1)

To verify the effectiveness of our HD-KT framework, we conducted future students' performance prediction experiments in the above four datasets. Table 2 shows the experimental results of the proposed HD-KT implemented in three KTMs compared with the baselines. First, it is clear that integrating our HD-KT framework to filter out the anomalous learning interaction has resulted in marked improvements in the performance of various KT models on all the

**Table 2: The overall performance comparison on four real-world datasets. The best results are shown in bold. All improvements are statistically significant (i.e., two-sided t-test with $p<0.01$).**

| Datasets | ASSISTment12 | | ASSISTment17 | | Slepemapy.cz | | Junyi | |
|---|---|---|---|---|---|---|---|---|
| Metrics | ACC | AUC | ACC | AUC | ACC | AUC | ACC | AUC |
| DTransformer | 0.7484±.001 | 0.7650±.001 | 0.7008±.001 | 0.7350±.002 | 0.8030±.001 | 0.7559±.002 | 0.8485±.002 | 0.7772±.001 |
| DKT | 0.7205±.002 | 0.6836±.001 | 0.6671±.001 | 0.6816±.001 | 0.7946±.001 | 0.6898±.002 | 0.8366±.002 | 0.7023±.001 |
| DKT+DSAN | 0.7211±.001 | 0.6845±.001 | 0.6679±.001 | 0.6821±.003 | 0.7967±.001 | 0.6915±.001 | 0.8368±.001 | 0.7054±.001 |
| DKT+GDN | 0.7236±.002 | 0.6863±.001 | 0.6695±.002 | 0.6847±.001 | 0.7973±.001 | 0.6930±.001 | 0.8395±.002 | 0.7063±.001 |
| HD-DKT | 0.7258±.001 | 0.6895±.001 | 0.6709±.001 | 0.6857±.001 | 0.8003±.002 | 0.6979±.003 | 0.8424±.001 | 0.7115±.002 |
| HawkesKT | 0.7441±.001 | 0.7559±.002 | 0.6845±.001 | 0.7033±.001 | 0.8058±.001 | 0.7574±.001 | 0.8410±.001 | 0.7609±.001 |
| HawkesKT+DSAN | 0.7449±.002 | 0.7604±.001 | 0.6865±.002 | 0.7078±.001 | 0.8064±.001 | 0.7608±.001 | 0.8430±.002 | 0.7634±.001 |
| HawkesKT+GDN | 0.7462±.001 | 0.7612±.001 | 0.6880±.001 | 0.7136±.001 | 0.8073±.002 | 0.7662±.002 | 0.8436±.001 | 0.7637±.001 |
| HD-HawkesKT | 0.7493±.002 | 0.7653±.002 | 0.6921±.002 | 0.7194±.001 | 0.8097±.001 | 0.7685±.003 | 0.8464±.001 | 0.7683±.002 |
| LPKT | 0.7541±.002 | 0.7750±.001 | 0.7172±.001 | 0.7635±.001 | 0.8061±.002 | 0.7648±.002 | 0.8517±.001 | 0.7926±.002 |
| LPKT+DSAN | 0.7577±.001 | 0.7764±.001 | 0.7188±.001 | 0.7670±.001 | 0.8096±.001 | 0.7681±.002 | 0.8527±.001 | 0.7931±.001 |
| LPKT+GDN | 0.7585±.001 | 0.7771±.001 | 0.7193±.001 | 0.7677±.002 | 0.8113±.001 | 0.7709±.001 | 0.8540±.002 | 0.7974±.001 |
| HD-LPKT | **0.7632±.001** | **0.7797±.001** | **0.7231±.002** | **0.7714±.001** | **0.8164±.001** | **0.7768±.001** | **0.8581±.001** | **0.8009±.001** |

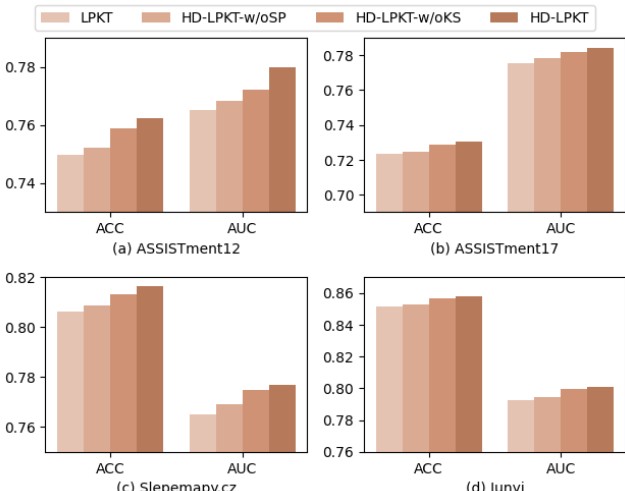

**Figure 4: The performance comparison of different modules.**

datasets. Compared to the original LPKT, the performance improvements of HD-LPKT on four datasets in terms of ACC as well as AUC are 1.21%, 0.82%, 1.28%, and 0.76% as well as 0.61%, 1.04%, 1.56%, and 1.04%, respectively. Second, we observed that integrating either DSAN or GDN with various KTMs can also effectively enhance the model's performance. However, compared to these variations, our method consistently achieved superior results. This further demonstrates the effectiveness of our approach in identifying anomalous learning interactions. Third, compared to the state-of-the-art robust KT model, i.e., DTransformer, our HD-LPKT model outperformed them on all four datasets. Indeed, rather than modifying the KTM network structure, our approach offers greater flexibility, allowing it to adapt to various KTMs.

## 5.3 Ablation Study (RQ2)

To answer RQ2, we conducted ablation experiments to investigate the effectiveness of our knowledge state-guided anomaly detector and student profile-guided anomaly detector. Due to the limited space, we compared HD-LPKT with **HD-LPKT-w/oKS** and **HD-LPKT-w/oSP**, which denote the variants of HD-LPKT without

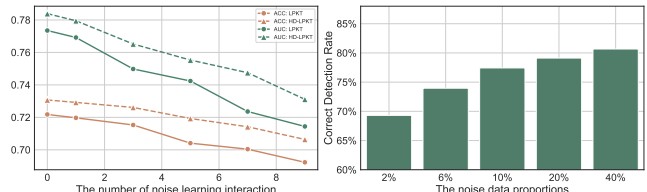

**Figure 5: The HD-KT's performance on simulated noise. (a) Left: The performance of LPKT and HD-LPKT with different numbers of noise data. (b) Right: The correct anomaly detection rate under different noise data proportions.**

knowledge state-guided and student profile-guided anomaly detectors, respectively. Figure 4 shows the performance comparison on ASSISTment12, ASSISTment17, Slepemapy.cz, and Junyi datasets. Clearly, removing any module will diminish the LPKT's performance. Notably, the removal of the student profile-guided anomaly detector has a more pronounced impact. However, retaining just one of the anomaly detectors can still reduce noise in learning interactions and enhance the effectiveness of the KTM.

## 5.4 Case Study (RQ1 & RQ3)

*5.4.1 The HD-KT's Performance on Simulated Noise.* Given that we do not have actual labels for anomalous learning interactions, to further validate the effectiveness of our method under noisy data conditions, we constructed simulated anomalous learning interaction data on the ASSISTment17 dataset for additional verification. Specifically, we randomly reversed the interaction data for each student. That is, if the original $r_t$ was 1, we changed it to 0, and vice versa. The left side of Figure 5 illustrates the performance difference between our HD-LPKT model and the original LPKT after introducing varying numbers of noise data points for each student. We can observe that as the amount of noise data increases, the performance of both models declines. However, the decline is more gradual for HD-LPKT, thereby validating that our model is more robust compared to the original KTM. It is noteworthy that even without adding any simulated noise data, our model still outperforms, as it can capture potential anomalies present in the

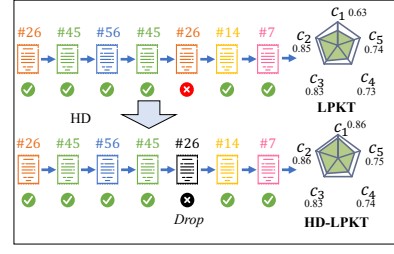

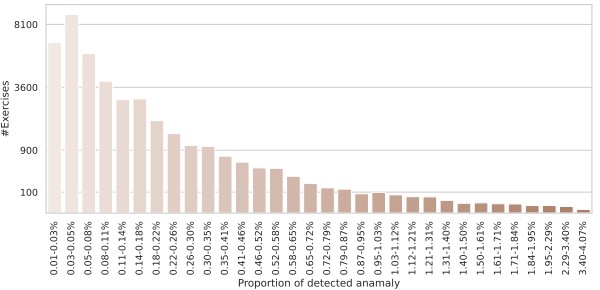

| Knowledge Concepts | Exercises |
|---|---|
| C1:Isosceles Triangle | #26,#18 |
| C2:Subtraction | #45,#266 |
| C3:Area | #56 |
| C4:Perimeter | #14 |
| C5:Equation Solving | #7 |
| C6:Square | #41 |
| C7:Addition | #13 |

| Models | #18 | #41 | #266 | #13 |
|---|---|---|---|---|
| LPKT | 0 | 0 | 1 | 0 |
| HD-LPKT | 1 | 1 | 1 | 1 |
| Truth | 1 | 1 | 1 | 1 |

Figure 6: A case study of HD-LPKT. This experiment shows the knowledge proficiency radar chart of the student with ID #14 in the Junyi data set using LPKT or HD-LPKT.

Figure 7: Distribution of anomalous proportions for exercises in ASSISTment17's learning interactions.

original dataset. Additionally, the right side of Figure 5 presents the proportion of noise data correctly identified by our HD-LPKT after introducing varying percentages of noise data into each student's learning sequence. It can be observed that even added 2% noise data, our framework can still accurately capture approximately 70% of them. Moreover, as the amount of noise data increases, the effective detection rate of our model gradually rises. This is because the more noise introduced, the more volatile the student's knowledge state becomes, making it easier for our model to detect.

*5.4.2 One Case Study of KT on Junyi Dataset.* In this case study, we showcased the results of knowledge tracing for student #14's learning sequence in the Junyi dataset using both HD-LPKT and LPKT. The results are presented in Figure 6. We can observe that our HD-LPKT model identified the second interaction with exercise #26 as anomalous, leading to the HD-LPKT and LPKT models diagnosing the student's mastery level of the knowledge concept "$c_1$: Isoseles triangle" as 0.86 and 0.63, respectively. Subsequently, we found that for future answer predictions related to exercise #18, which is associated with the knowledge concept $c_1$, our HD-LPKT model could predict accurately, while LPKT could not. Moreover, for the knowledge concept "$c_6$: Square", which potentially relates to $c_1$, our model also predicts the student's future performance more effectively. This validates that our HD-KT framework can robustly diagnose students' knowledge states by removing anomalous data from learning interactions.

*5.4.3 Exercise Quality Analysis.* In online learning systems, an important task is to evaluate the quality of exercises, since high-quality exercises can more precisely track the students' knowledge states. Our method enables the detection of anomalous learning interactions within the data, facilitating an analysis of the proportion of

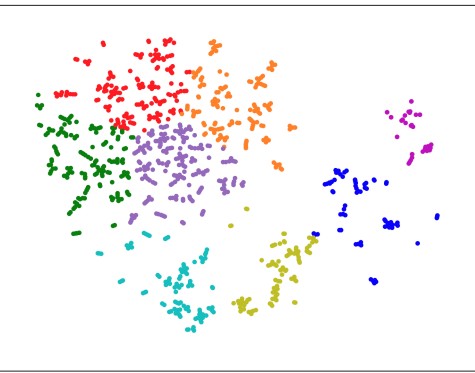

Figure 8: Student clustering based on the proportion of detected anomalous interactions, wherein we sampled 1000 students in ASSISTment12. We used K-means to cluster the students and marked them with different colors accordingly.

anomalies across various exercises during data interaction. Figure 7 illustrates the distribution of exercise across different anomaly proportions in ASSISTment17. This result can serve as a basis for exercise quality analysis, whereby exercises detected with a higher anomaly rate can be revisited and reviewed by domain experts.

*5.4.4 Learning Behavior-based Student Clustering.* As previously mentioned, some anomalous interactions in a student's learning sequence result from their learning behavior, such as carelessness. Here, we identify student groups with similar learning behaviors by analyzing the detected anomalous interactions from our HD-KT. Specifically, we first computed the proportion of detected anomalous interactions per student, for each knowledge concept, relative to all interactions associated with that concept. Subsequently, we utilized these proportions as feature vectors, representing potential anomalous behaviors of students across various knowledge concepts. These vectors were visualized after dimensionality reduction using t-distributed stochastic neighbor embedding (t-SNE). As shown in Figure 8, students with similar anomalous behavior are grouped into distinct clusters. These separated student groups assist educators in identifying representative student behavior patterns, enabling the creation of more tailored online learning experiences.

## 6 CONCLUSION

In this paper, we proposed a novel framework, termed **HD-KT**, to enhance the robustness of existing knowledge tracing (**KT**) methodologies with **H**ybrid learning interactions **D**enoising approach. In HD-KT, two detectors for anomalous learning interactions (namely knowledge state-guided anomaly detector and student profile-guided anomaly detector) were specially designed. More specifically, in the first detection module, a sequential autoencoder was designed to identify anomalous learning interactions by detecting atypical student knowledge states. In the second module, an attention mechanism was incorporated by modeling a student's long-term profile to capture irregular interactions. Extensive experiments validate the significant advantages of our HD-KT from multiple aspects. On the one hand, HD-KT markedly boosts both the robustness and accuracy of prevailing KT models. On the other hand, the HD-KT can facilitate exercise quality analysis and learning behavior-based student clustering.

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
