# OpenReview forum: "HD-KT: Advancing Robust Knowledge Tracing via Anomalous Learning Interaction Detection"
_ACM.org/TheWebConf/2024/Conference — TheWebConf24 Oral_

### Official Review · Reviewer_B9KM · 2023-11-29

**Novelty:** 6
**Technical Quality:** 6

**Review:**

This paper studies the problem of Knowledge Tracing (KT) with the goal to trace and predict each student’s knowledge states throughout their learning process. KT is an important part for online learning. Different from previous work that focus on revising neural network architecture  to improve KT performances, this paper takes a different angle by checking the anomalous interactions in students’ learning processes. It then proposes a method by introducing two anomalous detectors, one to detect anomalous interactions guided by knowledge state, and the other to detect anomalous interactions based on student profiles. Experiments on four real-world benchmark datasets demonstrate that the proposed method is robust to anomalous interactions, and outperforms baselines in predicting future student performances. The paper is well motivated and is easy to follow.

This paper enhance the robustness of KT by explicitly introducing two anomalous detectors. How does the proposed method compared to methods that apply adversarial training to enhance knowledge tracing (e.g., ref [10])?

**Questions:**

1. How does the proposed method compared to methods that apply adversarial training to enhance knowledge tracing (e.g., ref [10])?

**Reviewer Confidence:**

1: The reviewer's evaluation is an educated guess

**Scope:**

3: The work is somewhat relevant to the Web and to the track, and is of narrow interest to a sub-community

---

### Official Review · Reviewer_c4gc · 2023-11-30

**Novelty:** 6
**Technical Quality:** 5

**Review:**

The paper presents a methodology to improve knowledge tracing (KT) by taking into account the case where an anomalous observation might exist in the data. More specifically, the paper proposes a pre-processing step, to correct any anomalies detected, so that any KT model could be used with this. Four different TK methods were tested as the subsequent model, along with other competing approaches. The results support the claims of the paper.

While the individual ideas are not something new, their application in the proposed way for KT is very interesting. The paper motivates and discusses the motivation behind the approach. In order to evaluate datasets that explicitly have outliers, a case study is also conducted by introducing noise in one dataset. The experimental setup is clear.

On the other hand, there are some sections of the paper that could be reduced in size. The experiment in Sect. 5.4.3 is rushed, without having a good explanation. Fig. 5 (and the left figure in particular) could be a bit bigger. Some sentences are very lengthy (e.g. lines 338-344).

**Questions:**

None.

**Reviewer Confidence:**

3: The reviewer is confident but not certain that the evaluation is correct

**Scope:**

4: The work is relevant to the Web and to the track, and is of broad interest to the community

---

### Official Review · Reviewer_xrnc · 2023-11-30

**Novelty:** 4
**Technical Quality:** 5

**Review:**

The authors propose a novel framework, HD-KT, to enhance performance of existing Knowledge Tracing Models (KTMs) by means of an approach that can reduce the impact of noisy interactions. The proposed solution involves the creation of representations for students, exercises, and concepts based on the student's learning sequence, with the addition of two detectors for identifying anomalous learning interactions. Experiments on four datasets show that the proposed approach can enhance the overall performance of KT models, with also some case study examples.

Strengths
+ Very well-written and well-structured, the authors conveyed their idea in a clear and concise way.
+ The method appears sound and convincing, with both conceptual and technical descriptions.
+ The experiments cover four datasets, which allows to show that the proposed framework can generalize.

Limitations
- Figure Descriptions: While the figures appear well-elaborated yet complex, their descriptions are limited, limiting the reader's ability to grasp the main messages they convey. It is recommended that the authors should extend the captions, particularly in Figure 1 and Figure 4, to provide a more comprehensive understanding of the visual content.
- Clarity on Abnormal Interactions: The paper focuses on detecting abnormal interactions, yet lacks a clear definition or an extensive exploratory analysis to concretely illustrate what abnormal means in the context presented. A more defined conceptualization and thorough exploration of abnormal interactions would enhance the clarity of the paper.
- Motivation for Detectors: In the method sections, the authors introduce two detectors for anomalous interactions, but there is insufficient motivation explaining why these specific detectors were chosen and why both are necessary. Providing a robust rationale for the selection of these detectors would strengthen the foundation of the proposed framework.
- Source Code Sharing: Although the paper covers various datasets and baselines with detailed implementation information, the absence of shared source code may pose challenges for other researchers attempting to reproduce the work, given also that some baselines were re-created from scratch. Sharing the source code would contribute to the reproducibility.
- Precision of Improvement: The experimental results highlight the framework's outperformance of several baselines, as indicated in Table 2. However, the reported gains, often at the second or third decimal, leave room for ambiguity regarding the real impact on students' learning. To show the significance of the improvement, it is suggested that the authors consider an online evaluation with A/B testing to provide a more practical context for their findings.
- Comprehensive Error Analysis: While the paper includes a robustness demonstration on simulated noise (Section 5.4.1) and a case study (Section 4.5.2), a more extensive error analysis is recommended. Providing insights into the frequency of cases where the proposed framework provides a positive impact within the datasets would further fortify the paper's claims and contribute to its overall robustness. Furthermore, the decisions about the way the noise is simulated should be better motivated, as the proposed one appears too much simplistic.

Overall, addressing these points would enhance the clarity, motivation, reproducibility, and practical implications of the proposed framework, strengthening the paper for publication.

**Questions:**

- Besides the shown experiments, why do you thing that both these detectors are necessary, conceptually speaking?
- Have you considered running an online evaluation with A/B testing to provide a more practical demonstration of gains?
- How the depicted case studies have been selected? What is the motivation behind that noise simulation?

**Reviewer Confidence:**

4: The reviewer is certain that the evaluation is correct and very familiar with the relevant literature

**Scope:**

3: The work is somewhat relevant to the Web and to the track, and is of narrow interest to a sub-community

---

### Official Review · Reviewer_Cbef · 2023-11-30

**Novelty:** 5
**Technical Quality:** 6

**Review:**

**Summary**

The authors introduce a model-agnostic framework, Hybrid Learning Interactions Denoising Knowledge Tracing (HD-KT), designed for sequential neural networks. The main contributions of this framework include the modeling of anomalies in knowledge states (e.g., carelessness) and student profiles (e.g., varying characteristics among students), specifically through Knowledge State-Guided Anomaly Detector and Student Profile-Guided Anomaly Detector. Through comprehensive experiments, they demonstrate the effectiveness of their proposed framework across three distinct sequential neural networks, namely DKT, HawkesKT, and LPKT.

**Strengths**

1. **Comprehensive Technical Coverage**. The authors cover and explain the necessary technical intricacies within their proposed framework, facilitating easy understanding and high level of reproducibility. The well-constructed anomaly components are also substantiated with deliberate intent.

2. **Elaborated Experiments & Promising Experiment Results**. The authors apply their framework to three models, comparing them with several state-of-the-art baseline models. The obtained performance and ablation studies highlight the promising capabilities of the proposed framework, as evidenced by performance improvements and low standard deviations.

3. **Facilities Additional Analysis**. The authors illustrate potential applications of anomaly components in analyzing exercise quality and in clustering students based on learning behavior.

**Weakness**

1. Classification of Anomalies. Although the authors have outlined certain common knowledge-state anomalies (e.g., carelessness), it is unclear what are the other possible knowledge-state anomalies and student profile-guided anomalies. Given that the primary contributions of this paper revolve around anomaly modeling, it would be advantageous for the authors to provide (1) additional instances of real-world anomalies and (2) illustrate how the proposed framework is proficient in capturing and facilitating the analysis of these diverse anomalies.

I have read the author's rebuttal, and their comments have clarified the questions I had.

**Questions:**

1. What are the other classification of anomalies present in knowledge tracing?
2.  Is the proposed framework able to capture these other classification of anomalies (if applicable)?
3. [Typo] **Line 334**. If I am not mistaken, "combining the knowledge concept $c_t$ ..." should be "combining the knowledge concept $k_t$ ..."

**Reviewer Confidence:**

3: The reviewer is confident but not certain that the evaluation is correct

**Scope:**

4: The work is relevant to the Web and to the track, and is of broad interest to the community

---

### Official Review · Reviewer_7q2H · 2023-12-01

**Novelty:** 6
**Technical Quality:** 6

**Review:**

**Summary**

This paper proposed HD-KT, a robust framework for Knowledge Tracing for mitigating abnormal interactions. HD-KT consists of two detectors: knowledge state-guided and student profile-guided anomaly detectors to identify anomalous learning interactions and to capture students’ irregular interactions. This paper extensively evaluates the proposed framework with various datasets and demonstrates its effectiveness in downstream tasks through diverse case studies.


**Strength**
* The paper is well-structured and easy to follow, clearly presenting the rationale behind model design choices and evaluation results.

* The problem of mitigating abnormal interactions is well-motivated. The paper proves the contribution of the model by solving difficult and important problems.

* The model design reflects careful consideration, with different aspects of abnormal interactions and corresponding appropriate desgin choices made for the goal of detectors.
* The paper conducts thorough experiments to evaluate the model, contributing to a comprehensive understanding of its model across various scenarios.

**Weaknesses**
* While the proposed model leverages advanced machine learning techniques, the performance of the model is marginal. In Table 2, the proposed method shows marginal performance increases compared to baselines (e.g., DKT, HawkesKT, LPKT), despite consistently outperforming them.  Additionally, Figure 5 indicates that the gap between the baseline and HD+ baseline increases slightly when the number of noise learning interactions increases.
* The purpose of Section 5.4.4 Learning behavior-based student clustering is not clearly understood. Providing more context or explanation for this section's necessity would be helpful in terms of evaluation.

**Questions:**

* Why did the authors not use the benchmark datasets used in the paper proposing DTransformer?
* The paper does not provide the source code. Are the authors willing to release the source code if the paper is accepted?

**Typo**
* …s set of… should be corrected to ...a set of.... in the  267th line of Section 3.
* In line 479 of Section 4.2, a period is used instead of a comma.

**Reviewer Confidence:**

2: The reviewer is willing to defend the evaluation, but it is likely that the reviewer did not understand parts of the paper

**Scope:**

4: The work is relevant to the Web and to the track, and is of broad interest to the community

---

### Official Review · Reviewer_MrcC · 2023-12-01

**Novelty:** 3
**Technical Quality:** 3

**Review:**

To address anomalous learning interaction, the authors propose a novel framework(HD-KT), which contains of knowledge state-guided anomaly detector and student profile-guided anomaly detector, to enhance the robustness of existing knowledge tracing (KT) methodologies. The problem to be solved in the paper is interesting, but the method proposed in the article is not innovative enough.

Advantages:
1. The paper is well-structured and the presentation, and chart are easy to follow.
2. The experiments are relatively complete, including performance comparison, ablation study and case study.
3. Studying anomaly detection in Knowledge tracing is a meaningful topic.

Weakness:
1.The method proposed in this article is not technically innovative. For example, the Knowledge State-Guided Anomaly Detector is simply composed of VAE and CNN.
2.The article's explanation of the method is not sufficient and reasonable. For example, in Section 4.2, why VAE and CNN can be used for anomaly detection?

**Questions:**

See the weekness.

**Reviewer Confidence:**

3: The reviewer is confident but not certain that the evaluation is correct

**Scope:**

4: The work is relevant to the Web and to the track, and is of broad interest to the community

---

### Official Review · Reviewer_Qimr · 2023-12-10

**Novelty:** 4
**Technical Quality:** 4

**Review:**

This work proposes a new framework called HD-KT to enhance the robustness of existing Knowledge Tracing Models (KTMs) in online learning. The framework incorporates two detectors for identifying anomalous learning interactions: a knowledge state-guided anomaly detector and a student profile-guided anomaly detector. By detecting and addressing anomalous interactions caused by low data quality and abnormal student behaviors, HD-KT improves the accuracy of future student performance predictions. Experimental results on real-world datasets demonstrate the effectiveness of HD-KT in boosting the robustness of prevailing KTMs.

While the work on "HD-KT: Advancing Robust Knowledge Tracing via Anomalous Learning Interaction Detection" presents a novel framework for enhancing Knowledge Tracing Models (KTMs), there are a few potential weaknesses that could be addressed.

Firstly, the paper lacks a detailed discussion on the limitations and potential challenges of implementing the proposed HD-KT framework in real-world educational settings. It would be beneficial to provide insights into the scalability, computational requirements, and potential ethical considerations associated with deploying the framework in practical online learning environments.

Secondly, the paper could benefit from a more comprehensive evaluation of the proposed HD-KT framework. While the experiments on four real-world benchmark datasets demonstrate its effectiveness, it would be valuable to include comparative studies with other state-of-the-art KTMs to better understand the relative performance and advantages of HD-KT.

Additionally, the paper could provide more concrete guidance or suggestions on how educators and institutions can integrate the HD-KT framework into existing online learning systems. This could involve discussing potential implementation strategies, addressing any potential barriers to adoption, and providing examples of successful integration in educational settings.

Discussing detailed parameter setting information of the new proposed method and comparing it with baselines is important. It allows for the reproducibility and transparency of the research. By providing specific parameter values and configurations, other researchers can replicate the experiments and validate the results.

**Questions:**

The paper "HD-KT: Advancing Robust Knowledge Tracing via Anomalous Learning Interaction Detection" could be improved in a few areas. Firstly, it should discuss the challenges of implementing the proposed framework in real-world educational settings, including scalability, computational requirements, and ethical considerations. Secondly, a more comprehensive evaluation should be conducted, comparing HD-KT with other state-of-the-art models to understand its relative performance. Additionally, the paper should provide guidance on integrating HD-KT into existing systems, addressing implementation strategies and potential barriers. Lastly, including detailed parameter settings and comparisons with baselines would enhance reproducibility and transparency.

**Reviewer Confidence:**

2: The reviewer is willing to defend the evaluation, but it is likely that the reviewer did not understand parts of the paper

**Scope:**

3: The work is somewhat relevant to the Web and to the track, and is of narrow interest to a sub-community

---

### Decision · Program_Chairs · 2024-01-22

**Decision:**

Accept (Oral)

**Comment:**

We received 7 solid reviews (including 2 emergent reviews) and appreciated the authors providing timely and detailed rebuttal responses to all reviewers. Reviewers are in consensus about the paper's fit for the conference and mostly agree with the novelty and quality of the work.

 The paper adds to the literature on knowledge tracing in online learning by considering the anomaly (a situation where the student's answering behavior on the exercise deviates from the student's knowledge state), and reviewers agree that identifying such anomaly is important in advancing knowledge tracing. The comprehensive evaluation (including additional evaluation and user study in rebuttal) shows consistent improvement of the proposed framework. Reviewers also noted that the proposed framework is model-agnostic and applied to different KT models.

 Most of the reviewers' questions have been addressed through the rebuttal process. The only unresolved concern is that several reviewers commented on the performance improvement being small. Yet it is worth noting that performance improvement should not be the only indicator of the contribution. The performance improvement may also be better perceived by the audience if the authors describe how the 1-2% increase translates into real-world impact.